# Maternal Steroids on Fetal Doppler Indices, in Growth-Restricted Fetuses with Abnormal Umbilical Flow from Pregnancies Complicated with Early-Onset Severe Preeclampsia

**DOI:** 10.3390/diagnostics13030428

**Published:** 2023-01-24

**Authors:** Oana Sorina Tica, Andrei Adrian Tica, Doriana Cojocaru, Irina Tica, Cristian Lucian Petcu, Victor Cojocaru, Dragos Ovidiu Alexandru, Vlad Iustin Tica

**Affiliations:** 1Department of “Mother and Child”, University of Medicine and Pharmacy of Craiova, 200349 Craiova, Romania; 2Craiova County Emergency Hospital, 200642 Craiova, Romania; 3Department of Pharmacology, University of Medicine and Pharmacy of Craiova, 200349 Craiova, Romania; 4Department of Anesthesiology and Intensive Care, “Nicolae Testemitanu” State University of Medicine and Pharmacy Chisinau, 2004 Chisinau, Moldova; 5“Timofei Mosneaga” Republican Clinical Hospital, 2025 Chisinau, Moldova; 6Department of Internal Medicine, Faculty of Medicine, University “Ovidius” Constanta, 900527 Constanța, Romania; 7Constanta County Emergency Hospital, 900591 Constanța, Romania; 8Department of Biophysics, Faculty of Dental Medicine, University “Ovidius” Constanta, 900527 Constanța, Romania; 9Department of Biostatistics, University of Medicine and Pharmacy of Craiova, 200349 Craiova, Romania; 10Department of Obstetrics and Gynecology, Faculty of Medicine, University “Ovidius” Constanta, 900527 Constanța, Romania

**Keywords:** umbilical artery, middle cerebral artery, cerebroplacental ratio, ductus venosus, uterine arteries, dexamethasone, pulsatility index, doppler fetal indices

## Abstract

Corticoids are largely used for fetal interest in expected preterm deliveries. This study went further, evaluating the effect of maternal administration of dexamethasone (Dex) on the umbilical artery (UA), middle cerebral artery (MCA), and ductus venous (DV) spectrum, in growth-restricted fetuses, with the absent end-diastolic flow (AEDF) in UA, from singleton early-onset severe preeclamptic pregnancies. Supplementary, the impact on both uterine arteries (UTAs) flow was also evaluated. In 68.7% of cases, the EDF was transiently restored (trAEDF group), in the rest of 31.2% remained persistent absent (prAEDF group). UA-PI significantly decreased in the first day after Dex (day 1/0; *p* < 0.05), reaching its minimum during day 2 (day 2/1; *p* > 0.05), revealing a significant recovery to day 4 (day 4/2; *p* < 0.05), in both groups. The MCA-PI decreased from day 1 until day 3 in both groups, but significantly only in the trAEDF group (*p* = 0.030 vs. *p* = 0.227. The DV-PI’s decrease (during day 1) and the CPR’s increase (between days 0 and 2) were not significant in both groups. UTAs-PIs did not vary. The prAEDF group had a significantly increased rate of antenatal worsening Doppler and a poorer perinatal outcome compared with the trAEDF group. In conclusion, Dex transiently restored the AEDF in UA in the majority of cases, a “positive” effect being a useful marker for better perinatal prognosis. UA-PI significantly decreased in all cases. The improvement in umbilical circulation probably was responsible for the short but not significant DV-PI reduction. MCA-PI decreased only in sensitive cases, probably due to an already cerebral “full” vasodilation in the prAEDF group. Furthermore, the CPR’s nonsignificant improvement was the result of a stronger effect of Dex on UA-PI than on MCA-PI. Finally, despite the same etiology, it was only a weak correlation between the severity of the umbilical and uterine abnormal spectrum.

## 1. Introduction

The incidence of early-onset preeclampsia (ESP) (actually renamed as preeclampsia with severe features) is generally estimated as being <1% [1]. In a twenty years study performed in the USA (1980–2000), Ananth et al. [2] found a slight increase in the incidence of preeclampsia, from 3.4% to 3.7%, but a more than 3-fold increase in the ponder of severe forms (from 0.3% to 1.4%). This serious disease is estimated to be responsible for maternal mortality up to 1.2/100.000 [3], which corresponds to more than 50.000 deaths worldwide each year [4]. At the same time, it represents an important cause of deliveries at early gestation ages, with high perinatal morbidity and mortality due to (severe) prematurity [5], with preeclampsia accounting for 25–43% of such births [2]. Consequently, the management of a pregnancy complicated by ESP, including the decision of the timing of delivery, continues to be a serious clinical challenge. For this purpose, ultrasonography, respectively a complex Doppler evaluation of fetal circulation, represents, by far, the most important investigation [6,7]. Furthermore, the occurrence of absent end-diastolic flow (AEDF) or reversed absent end-diastolic flow (REDF) in the umbilical artery (UA) is a sign of supplementary severity [8,9,10]. 

Steroids are typically used to improve fetal outcomes in expected preterm labor [11,12]. Their effect on the fetal Doppler indices in normal or complicated pregnancies, by any cause, is, however, differently reported by authors [13,14,15,16].

The current work is focused on the effect of maternal administration of dexamethasone (Dex) on fetal circulation in (severely) restricted fetuses, revealing AEDF in UA, exclusively from ESP pregnancies. Complementary, a correlation between the Doppler spectrum in UA and in the right and/or left uterine arteries (r/lUTAs) was also evaluated.

## 2. Materials and Methods

This study was performed between January 2010 and June 2022 at the County Emergency Hospital of Craiova, Romania, the County Emergency Hospital of Constanta, Romania, and at “Timofei Mosneaga” Republican Clinical Hospital of Chisinau, Moldova. The three are tertiary care institutions. All cases belonged to the “normal” management category of the protocol of the facility, and no special or supplementary explorations were done. The patients were informed about the study design and agreed that all results for them and their newborns could be anonymized and used in the present study if the inclusion criteria were met.

The inclusion criteria referred to singleton ESP pregnancies with ultrasound-certified fetal growth restriction (FGR) and AEDF in UA. No patient with fetal malformations was included. Fetuses with REDF in UA and/or absent or reversed A-wave in ductus venosus (DV), according to our institutional guidelines, underwent a rapid corticoid cure, if possible, and were delivered in the subsequent next 24–48 h. This is concordant with Karsdrop et al. [17], who found perinatal mortality of up to 50% in REDF fetuses, and with Schwarze et al. [18], who reported high mortality at any gestational age, if absent and/or reversed A-wave is observed. The corresponding patients were not eligible for the present study. The exclusion criteria also referred to patients with diabetes mellitus, obesity, vascular and/or connective disorders, or those receiving steroids for other reasons.

The diagnosis of ESP was established on generally accepted criteria. These included increased blood pressure occurring between 20 and 34 weeks of gestation in a previously non-hypertensive woman, with systolic value > 160 mmHg and diastolic value > 120 mmHg at one measurement, or >110 mmHg measured at least twice with an intervening 6-h rest and associated proteinuria, defined as >300 mg/day. In patients with no proteinuria in addition to hypertension, ESP was diagnosed based on one or several signs of severity. These signs included oliguria (<500 mL/24 h), thrombocytopenia (<100.000/mm^3^), impaired liver function (liver enzymes twice the normal activities), progressive renal insufficiency, new cerebral or visual disturbances, or epigastric or right upper quadrant pain [19,20,21,22].

FGR (also termed intrauterine growth restriction, IUGR) was considered to conform with the recommendation of the ISUOG Practice Guidelines: the fetal abdominal circumference (AC) below the 5th percentile of the normal range for gestation [23]. Severely restricted fetuses were considered fetuses with AC < 3rd percentile for the GA.

All patients received similar appropriate antihypertensive therapy. Magnesium sulfate was used to prevent seizures and for neonatal neuroprotection.

For fetal therapy, Dex was administered according to our institutional guidelines for estimated preterm births. The Dex regimen consisted of four doses of 6 mg intramuscularly at 12-h intervals. 

Spectral Doppler of UA, middle cerebral artery (MCA), DV, and supplementary r/lUTAs-PIs, were performed at inclusion, before the first dose of Dex (day 0 or hour 0), and at 24, 48, 72, and 96 h after inclusion (days 1 to 4). Each fetus was transabdominal scanned by the same expert, with the fetus in quiescence and apnea. For each determination, at least 10 uniform wave spectra were recorded, and the pulsatility index (PI) was automatically calculated by the system.

The normal ranges for UA-PI, MCA-PI, and cerebroplacental ratio (CPR) conformed to the Fetal Medicine Foundation charts [24], for DV-PI conformed to Kessler et al. [25] and Para-Cordero et al. [26] and for r/lUTA-PI, conform the reports of Geipel et al., [27], respectively. Based on our institutional guidelines, fetal blood redistribution was considered when CPR was ≤1.

Data analyses were performed using SPSS Statistics 23 software (IBM). The procedures used included descriptive statistics, graphs, parametric statistical tests (one-way ANOVA test), and nonparametric statistical tests: chi-square test (OR) and Kruskal Wallis test, respectively. The significance level was 0.05, and the confidence interval (CI) was 95%.

## 3. Results 

Initially, we included 37 singleton pregnancies fulfilling the criteria. All these fetuses were severely growth-restricted. Five patients that delivered (spontaneously or induced) in the first 48 h after the first dose of Dex were excluded since the interval was considered too short for evaluation and analysis.

The mean age of the patients included was 22.1 ± 5.6 years (ranging from 15 to 41 years).

### 3.1. Doppler Findings on Umbilical Artery (UA)

Doppler examination of the UA revealed Dex transiently restored the EDF in the UA in 22 (68.7%) cases. These cases comprised the transiently restored (absent) end-diastolic flow group (trAEDF). The remaining 10 (31.2%) fetuses showed no variation and comprised the persistent absent end-diastolic flow group (prAEDF). The effect started in the first 24 h after the first dose of Dex and lasted up to day 4. 

The prAEDF group had a significantly lower gestational age (GA) at admission compared with the trAEDF group (28 weeks 5 days ± 10 days vs. 29 weeks 6 days ± 8 days; *f*-ratio = 4.92624; *p* = 0.034). The GA was calculated according to the first day of the last menstrual period (GA(LMP)) and confirmed (when the results were available) by the data from the first-trimester ultrasound examination.

The UA Doppler showed increased velocities with UA-PI above the 95th percentile in all patients.

UA-PI significantly decreased in both groups on day 1. The decrease was even highly statistically significant in the trAEDF group. UA-PI continued to decrease slightly, reaching its minimum on day 2, but this decrease was not significant. Excepting one patient belonging to the trAEDF group with uncontrolled maternal hypertension and one fetus belonging to the prAEDF group who revealed newly occurred REDF in UA, both cases imposing termination of pregnancy during the third day, in all other cases, UA-PI returned to the values prior to steroid administration until day 4. In the prAEDF group, another fetus had to be delivered due to spontaneous heart decelerations on the 4th day and consecutively, the fourth day was not quantified in that case. The UA-PI values on day 2 and 4 were significantly different in both groups. There was also a statistically significant difference between day 1 and day 4 in the trAEDF group (*p* = 0.002) but not in the prAEDF group (*p* = 0.055) (Table 1, Table 2 and Table 3; Figure 1). 

Until delivery, in total, five cases (50%) belonging to the prAEDF group and in three (13.6%) cases from the trAEDF group, respectively, the Doppler worsened, with the occurrence of REDF in UA and/or absent A-wave in DV, imposing fetal extraction. The difference was statistically significant (OR: 6.3; *p* = 0.037).

### 3.2. Doppler Findings on Middle Cerebral Artery (MCA) and Cerebroplacental Ratio (CPR) Evaluation

Doppler examination of the MCA revealed that, prior to Dex administration, MCA-PI below the 5th percentile was evident in 5/10 (50.0%) of fetuses in the prAEDF group and in 6/22 cases (27.2%) in the trAEDF group (Figure 2). The difference was not significant (OR: 2.6; *p* = 0.216). Dex transiently decreased the MCA-PI in both groups, with a maximal effect on day 1 and a gradual return to values prior to corticoid treatment by 3 and 4. This variation was statistically significant only in the trAEDF group (*p* = 0.030) but not in the prAEDF group (*p* = 0.227) (Table 1, Table 2 and Table 4; Figure 2).

CPR was subunitary in all fetuses at recruitment (Table 5; Figure 3). 

CPR slightly increased from day 0 to day 2, followed by a gradual return to the initial value in the ensuing few days. The variation was significant in neither group (Table 1). 

### 3.3. Doppler Findings on Ductus Venosus (DV)

The DV Doppler revealed a DV-PI above the 95th percentile in 9/10 (90.0%) cases from the prAEDF group and in 15/22 (68.1%) cases from the trAEDF groups (Table 6). The difference between the groups was not significant (OR: 4.2; *p* = 0.211). 

DV-PI decreased slightly on the first day after Dex administration and recovered during day 2. This variation was significant in none of the groups (Table 1; Figure 4).

### 3.4. Doppler Findings on Right and Left Uterine Arteries (r/lUTA)

At admission, the PI was above the 95th percentile in both UTAs in all patients. UTAs-PIs showed no significant variation during the 5 days of observation in both groups (Table 1).

There were no differences between either rUTA-PI or lUTA-PI in the two groups (tr/prAEDF groups) (Figure 5).

## 4. Discussion

Our institutional protocols recommend Dex in expected preterm birth. No difference in efficacy between this steroid and betamethasone efficiency has been evident, and the selection depends on the clinical experience of the physician [28,29]. However, Senat et al. [13] and Muller et al. [14] reported a higher impact on the fetal variability rate and cerebral circulation induced by betamethasone when compared to Dex.

UA Doppler is the sole approach that offers both diagnostic and prognostic information for the management of FGR [6]. Doppler waves in the UA begin to be altered when approximately one-third of the placental villous is altered. More severe aspects, such as absence and REDF, occur when 60–70% of the villi are abnormal [30,31]. So, it is not surprising that all fetuses included were severely restricted. The occurrence of AEDF in UA is an important sign of fetal distress [32], with perinatal mortality being around 30% for AEDF and as high as 70% for reversed end diastolic flow—REDF [8,17,33].

In the current study, the UA-PI exceeded the 95th percentile in all AEDF cases, with high UA-PI being an early sign of severely increased resistance in UA [30,31].

Dex transiently restored the end-diastolic flow in UA in 68.7% of cases, a rate concordant with the reports of Smichen et al.—55% [34], Robertson et al.—63% [35] and Ekin et al.—79% [16], but significantly greater than data communicated by Muller et al.—45% [14].

The GA(LMP) was significantly lower in the prAEDF group compared with the trAEDF group because the maternal symptoms, such as: hypertension, headache, edema, abdominal pain, and visual disturbances, occurred earlier in gestation with an earlier diagnosis of EPS. The fact was probably, the result of more severe systemic abnormalities. Furthermore, these systemic abnormalities, possibly were the result of more defective placentation, with maternal impact but also with more severe fetal impact, revealed by UA non-response to Dex administration. This conclusion is concordant with the report of Moldenhauser et al. [36], who found that placental abnormalities are more severe when the disease occurs in early gestation. 

The mechanism of transiently restoring from absent (or reversed) to positive EDF in UA remains gloomy. It was proposed that steroids act through the placental corticotrophin-releasing hormone (CRH) dependent processes. Normally, steroids decrease the hypothalamic secretion of CRH but increase the placental production of the peptide [37]. The peptide binds specific receptors—CRH-R2, located in the placenta but also in umbilical vasculature [38], followed by increasing the expression of nitric oxide synthase and, consecutively, increased production of nitric oxide, with intense vasodilator effect [39]. However, this effect is present only in restricted fetuses and absent in “normal” ones, probably due to already fully activated nitric oxide synthase in the last group, compared with a deficient expression in FGR [16,40]. Hampl et al. [41] found that the increased resistivity in UA circulation is mainly the result of the (hypoxic) inhibition of muscular membrane K^+^-channels. So, it can be speculated a positive effect of dexamethasone on the expression of these channels, followed by vasodilatation.

The lack of an effect of Dex in prAEDF fetuses was a good predictor for a significantly increased rate of proximal fetal deterioration (including worsening Doppler and/or occurrence of spontaneous decelerations) and for a poorer neonatal outcome, especially involving hypoxia and hypoxia-induced complications [42].

Dex induced significant variation in UA-PI between days 1 and 0 and days 4 and 2 in both groups, but between days 4 and 1 only in the trAEDF group. It is likely that this last variation would have also been significant in the prAEDF group if this group comprised a larger cohort since the *p*-value of 0.055 was close to the cutoff value of 0.05.

We found an almost two-fold increase in the incidence of MCA-PI below the 5th percentile in the prAEDF group, compared with the trAEDF group. The difference was not significant. However, CPR was subunitary in all AEDF fetuses. The latter finding provides further evidence that CPR is the main instrument to evaluate fetal cerebral redistribution, but the cutoff value of CPR for poor perinatal outcome prognosis has been variously reported [43,44,45,46]. 

Dex decreased the MCA-PI during the first day after administration in both groups. This decrease was significant only in the trAEDF group. The difference was probably due to an already higher grade of vasodilatation in the fetal cerebral circulation in fetuses belonging to the prAEDF group. The mechanism of the steroid-induced MCA-PI reduction probably included cerebral involvement [37].

The slight and non-significant increase in CPR during the first day after Dex was secondary to a more pronounced decrease in UA-PI, compared with MCA-PI, in the same period.

DV-PI exceeded the 95th percentile in 90.0% of cases from the prAEDF group and 68.1% of cases from the trAEDF group, the difference being non-significant. These findings indicate that the severely increased resistivity in UA, with UA-PI > 95th percentile (which characterizes AEDF fetuses), preceded and triggered the increased blood shunt through DV, as revealed by the high DV-PI [47]. The abnormal DV Doppler, a late but a very severe aspect of fetal distress [48], is an important marker in establishing the right timing of delivery [49].

The decrease in DV-PI on the first day after maternal administration of Dex was probably a consequence of the improvement in the UA flow induced by the steroid. However, the decrease was not statistically significant. 

UTAs-PIs were above the 95th percentile in all cases. This fact is concordant with the report of Sebire et al. [30], who remarked that abnormal UTAs Doppler could occur long before altered UA waveforms can be observed. Increased velocities in UTAs are the result of systemic endothelial damage induced by defective placentation [50,51]. 

The fact that the amplitude of both rUTA-PI and lUTA-PI were not different in trAEDF compared with the prAEDF group highlights the possibility that the defective placentation differently interests the placental-uterine interface versus the placental-umbilical interface.

Furthermore, the UTAs-PIs were not affected by the corticoid, data concordant with the reports of Senat et al. [13] and Ekin et al. [16]. A possible explanation why dexamethasone (and/or betamethasone) can induce vasodilatation in umbilical circulation in ESP complicated with FGR and not also in uterine arteries belongs to Zhou et al. [52], who reported a high expression of CRH –R2 receptors in umbilical endothelium and syncytiotrophoblast, with vasodilator effect (mediated by the increase in nitric oxide synthase expression—see above) and almost lack of them in uteroplacental circulation.

## 5. Conclusions

In (severely) growth-restricted fetuses with AEDF in UA, from early-onset severe preeclamptic pregnancies, the fetal monitoring and delivery are based on a complex Doppler investigation, including UA, MCA, and DV.

UA-PI > 95th percentile precedes and always accompanies all AEDF cases, being an earlier and a less severe sign of increased resistance in UA. 

Dex, administered for fetal lung maturation, transiently restores the end-diastolic flux in UA in the majority of cases and decreases the UA-PI, this effect lasting up to the 4th day. 

In all growth-restricted fetuses with AEDF in UA, from ESP pregnancies, the CPR is subunitary, as proof of already severely installed blood redistribution.

Dex significantly decreased the MCA-PI during the first day after maternal administration in all fetuses. The decrease was significant only in the trAEDF group, probably as a consequence of more intense cerebral vasodilation in cases with persistent AEDF in UA.

DV-PI > 95th percentile was noticed at similar rate, but not in all cases, in transiently restored and in persistent AEDF in UA fetuses, fact proving that it is a marker of high fetal distress, occurring later than severe abnormalities in UA. The DV Doppler is a very important marker for delivery timing. 

The no significant DV-PI’s transiently reduction (during the first day) is probably the result of an improvement in umbilical circulation after Dex administration.

Increased velocities in UTAs (UTAs-PIs > 95th percentile) were noticed in all patients, but, despite the same etiology, uterine abnormalities only partially corroborate with the grade of umbilical artery distress.

Dexamethasone has no impact on UTAs-PI. 

## Figures and Tables

**Figure 1 diagnostics-13-00428-f001:**
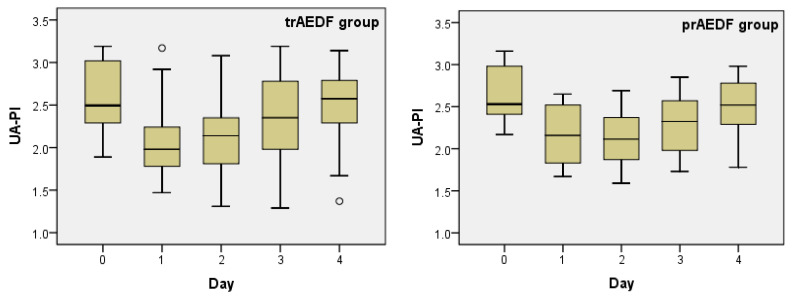
Boxplot charts on the pulsatility index’s evolution in the umbilical artery (UA-PI), prior (day 0) and in the first 4 days after maternal dexamethasone administration, in the two groups (tr/prAEDF = transiently restored/persistent absent end-diastolic flow in umbilical artery group).

**Figure 2 diagnostics-13-00428-f002:**
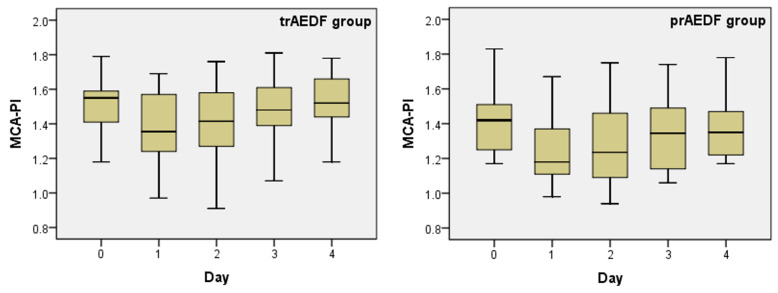
Boxplot charts on the pulsatility index’s evolution in the middle cerebral artery (MCA-PI), prior (day 0) and in the first 4 days after maternal dexamethasone administration, in the two groups (tr/prAEDF = transiently restored/persistent absent end-diastolic flow in umbilical artery group).

**Figure 3 diagnostics-13-00428-f003:**
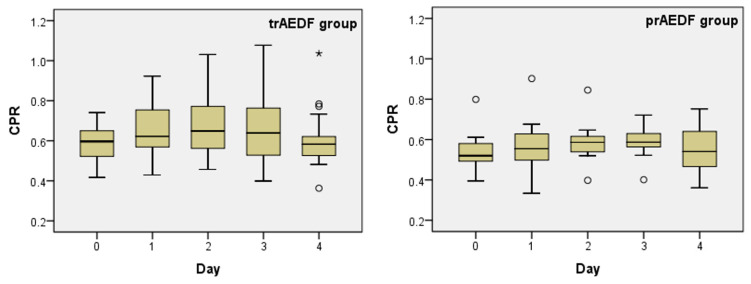
Boxplot charts on the cerebroplacental ratio (CPR) prior (day 0) and in the first 4 days after maternal dexamethasone administration in the two groups (tr/prAEDF = transiently restored/persistent absent end-diastolic flow in umbilical artery group).

**Figure 4 diagnostics-13-00428-f004:**
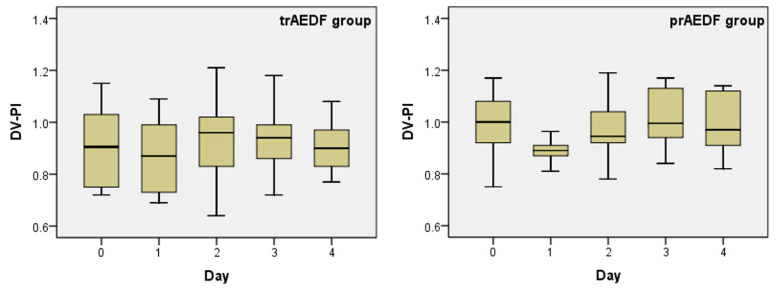
Boxplot charts on the pulsatility index in the ductus venosus (DV-PI), prior (day 0) and in the first 4 days after maternal dexamethasone administration, in the two groups (tr/prAEDF = transiently restored/persistent absent end-diastolic flow in umbilical artery group).

**Figure 5 diagnostics-13-00428-f005:**
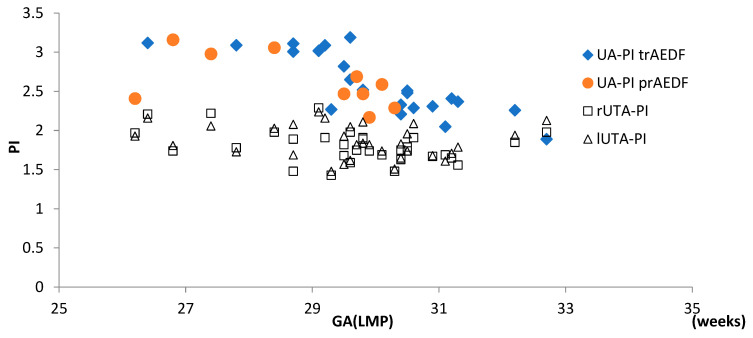
A complete image with the pulsatility indexes in the umbilical artery (UA-PI) and in the right and left uterine arteries (r/lUTA-PI), at admission, in the two groups. (tr/prAEDF = transiently restored/persistent absent end-diastolic flow in umbilical artery group).

**Table 1 diagnostics-13-00428-t001:** Kruskal-Wallis test results on the variation of the pulsatility indexes (PIs) in an umbilical artery (UA), middle cerebral artery (MCA), ductus venosus (DV), and right/left uterine arteries (r/lUTAs), and on the variation of cerebroplacental ratio (CPR), respectively, within the interval between the day 0 (prior) and in the first 4 days after maternal dexamethasone administration, in the two groups: transiently restored and persistent absent end-diastolic flow (tr/prAEDF) in UA. The test was significant (*p* < 0.05) for UA-PI in both groups and for MCA-PI in the trAEDF group. The test was nonsignificant (*p* > 0.05) for MCA-PI in the prAEDF group and for CPR, DV-PI, and r/lUTAs-PIs in both groups. (H = Statistic test value; d. f. = degree of freedom; *p* = value for the statistic test).

Kruskal-Wallis Test	UA-PI		MCA-PI		CPR		DV-PI		rUTA-PI		lUTA-PI	
	tr AEDF	pr AEDF	tr AEDF	pr AEDF	tr AEDF	pr AEDF	tr AEDF	pr AEDF	tr AEDF	pr AEDF	tr AEDF	pr AEDF
H	20.56	11.35	10.71	11.35	6.08	3.32	3.85	10.80	1.23	0.196	1.98	0.32
d. f.	4	4	4	4	4	4	4	4	4	4	4	4
*p*	0.000	0.023	0.030	0.227	0.193	0.506	0.426	0.395	0.872	0.995	0.738	0.988

**Table 2 diagnostics-13-00428-t002:** Post-hoc analysis for comparing the pulsatility indexes (PIs) registered on different days, between day 0 (prior) and the first 4 days after maternal dexamethasone administration, in the case of a significant (*p* < 0.05) Kruskal-Wallis test. The test was significant for UA-PI in both groups and for MCA-PI in the trAEDF group. (*) For MCA-PI in the prAEDF group, the Kruskal-Wallis test was nonsignificant (*p* > 0.227).

Parameter		Days Compared (*p*)									
		0/1	0/2	0/3	0/4	1/2	1/3	1/4	2/3	2/4	3/4
UA-PI	trAEDF	0.001	0.001	0.091	0.744	0.841	0.077	0.002	0.118	0.003	0.172
	prAEDF	0.010	0.006	0.079	0.500	0.896	0.403	0.055	0.334	0.041	0.055
MCA-PI	trAEDF	0.030	0.109	0.782	0.684	0.465	0.040	0.006	0.185	0.045	0.495
	prAEDF	(*)									

**Table 3 diagnostics-13-00428-t003:** The characteristics of the pulsatility index in the umbilical artery (UA-PI) prior (day 0) and in the first 4 days after maternal dexamethasone administration in the two groups. (tr/prAEDF = transiently restored/persistent absent end-diastolic flow in umbilical artery group).

UA-PI	Day									
	0		1		2		3		4	
	tr AEDF	pr AEDF	tr AEDF	pr AEDF	tr AEDF	pr AEDF	tr AEDF	pr AEDF	tr AEDF	Pr AEDF
Cases	22	10	22	10	22	10	22	10	22	10
Mean	2.591	2.629	2.110	2.145	2.097	2.133	2.312	2.303	2.512	2.494
Median	2.495	2.530	1.980	2.160	2.140	2.115	2.350	2.325	2.575	2.520
Std. Deviation	0.397	0.337	0.453	0.376	0.452	0.348	0.506	0.350	0.457	0.394
Minimum	1.890	2.170	1.470	1.670	1.310	1.590	1.290	1.730	1.370	1.780
Maximum	3.190	3.160	3.170	2.650	3.080	2.690	3.190	2.850	3.140	2.980

**Table 4 diagnostics-13-00428-t004:** The characteristics of the pulsatility index in the middle cerebral artery (MCA-PI) prior (day 0) and in the first 4 days after maternal dexamethasone administration in the two groups. (tr/prAEDF = transiently restored/persistent absent end-diastolic flow in umbilical artery group).

MCA-PI	Day									
	0		1		2		3		4	
	tr AEDF	pr AEDF	tr AEDF	pr AEDF	tr AEDF	pr AEDF	tr AEDF	pr AEDF	tr AEDF	pr AEDF
Cases	22	10	22	10	22	10	22	10	22	10
Mean	1.510	1.414	1.373	1.246	1.395	1.289	1.486	1.357	1.530	1.397
Median	1.550	1.420	1.355	1.180	1.415	1.235	1.480	1.345	1.520	1.350
Std. Deviation	0.146	0.200	0.192	0.203	0.233	0.242	0.180	0.220	0.145	0.219
Minimum	1.180	1.170	0.970	0.980	0.910	0.940	1.070	1.060	1.180	1.170
Maximum	1.790	1.830	1.690	1.670	1.760	1.750	1.810	1.740	1.780	1.780

**Table 5 diagnostics-13-00428-t005:** The characteristics of the cerebroplacental ratio (CPR) prior (day 0) and in the first 4 days after maternal dexamethasone administration in the two groups. (tr/prAEDF = transiently restored/persistent absent end-diastolic flow in umbilical artery group).

CPR	Day									
	0		1		2		3		4	
	tr AEDF	pr AEDF	tr AEDF	pr AEDF	tr AEDF	pr AEDF	tr AEDF	pr AEDF	tr AEDF	Pr AEDF
Cases	22	10	22	10	22	10	22	10	22	10
Mean	0.588	0.537	0.655	0.574	0.688	0.592	0.656	0.585	0.605	0.552
Median	0.597	0.520	0.623	0.555	0.649	0.587	0.639	0.587	0.583	0.541
Std. Deviation	0.080	0.115	0.120	0.149	0.171	0.112	0.180	0.087	0.137	0.125
Minimum	0.417	0.395	0.429	0.334	0.457	0.398	0.399	0.401	0.363	0.361
Maximum	0.741	0.799	0.923	0.902	1.031	0.845	1.077	0.721	1.036	0.752

**Table 6 diagnostics-13-00428-t006:** The pulsatility index in ductus venosus (DV-PI), prior (day 0) and in the first 4 days after maternal dexamethasone administration in the two groups. (tr/prAEDF = transiently restored/persistent absent end-diastolic flow in umbilical artery group).

DV-PI	Day									
	0		1		2		3		4	
	tr AEDF	pr AEDF	tr AEDF	pr AEDF	tr AEDF	pr AEDF	tr AEDF	pr AEDF	tr AEDF	Pr AEDF
Cases	22	10	22	10	22	10	22	10	22	10
Mean	0.912	1.000	0.874	0.894	0.944	0.968	0.942	1.025	0.905	1.001
Median	0.905	1.000	0.870	0.890	0.960	0.945	0.940	0.995	0.900	0.970
Std. Deviation	0.149	0.127	0.134	0.044	0.151	0.110	0.119	0.114	0.089	0.116
Minimum	0.720	0.750	0.690	0.810	0.640	0.780	0.720	0.840	0.770	0.820
Maximum	1.150	1.170	1.090	0.964	1.210	1.190	1.180	1.170	1.080	1.140

## Data Availability

Not applicable.

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
