# Peer review of "Maternal Steroids on Fetal Doppler Indices, in Growth-Restricted Fetuses with Abnormal Umbilical Flow from Pregnancies Complicated with Early-Onset Severe Preeclampsia"

_diagnostics, 2023, doi:10.3390/diagnostics13030428_

Round 1

Reviewer 1 Report

Tica et al. report a correlation study on the variation of UA-PI, MCA-PI, CPR, and DV-PI with the response of AEDF in UA to maternal administration of dexamethasone (Dex). This study was conducted in pregnancies diagnosed with early-onset severe preeclampsia and fetal growth restriction. The authors can distinguish two AEDF groups and the effect of Dex on each variable determined via Doppler: Most with the restored absence of flow and the rest with the persistence of absent flow.

This study aims to add new procedures (i.e., correlations) to interpret Doppler data and complement clinical data. However, several aspects must be improved.

Abstract

1. Although the abstract is well structured, it is suggested to add a general sentence that summarizes all the findings obtained to clarify the main ideas and conclusions of the study.

Introduction

1.  Line 18: The authors use published evidence to report that the effects of steroid use have been differentially reported. It is suggested to clarify how the study conducted by Tica et al. will deepen the current knowledge and relate it to the hypothesis or the main aim of this work.

Material and methods

1. It would be valuable to add a flow chart that summarizes the inclusion and exclusion criteria of the present study to facilitate the understanding of the criteria and to be clear about the number of individuals used in each analysis.

2. 4th paragraph: Although Fetal Growth Restriction (FGR) consideration is correct, it is suggested to cite a peer reviewed - published article in which FGR definition criteria be clearly explained.

3. 9th paragraph: The authors state the use of parametric ANOVA test, but this test was not clearly used in the results section.

4. The authors must add an explicit paragraph regarding that the ethical protocol was approved by Institutional Committee.

Results

General observation:

1.The number of individuals used in each reported analysis must be explicit in each of the tables and figures.

2. It is unclear in all figures if deviation in average values is Standard Deviation or Standard Error. Authors must clarify this point.

3.1

Both groups are characterized concerning gestational age (GA). The result is not explicitly associated with any figure or table. It is requested to add a representative figure or table.

Then, it is necessary to clarify which results belong to what is shown in table 1 (Comparing PI of different blood vessels), table 2 (comparison UA ​​and MCA PI on various days), and figure 1 (UA on different days and different groups) since the analysis in each one is different. It is requested that to reorder this section and include values ​​with standard error for each measurement.

3.2

In this section, the average values ​​with standard error associated with each of the measurements must be included.

3.3

In this section, the average values ​​with standard error associated with each of the measurements must be included.

Figures

It is suggested to make the Y-axis scale consistent in each of the figures (i.e., Figure 3 uses values ​​between 0.20 and 1.20, with intervals of 0.20 for the trAEDF group while in the trAEDF group prAEDF intervals of 0.10 and values ​​between 0.30 and 1.00 are used). In addition, it is suggested that the legend of the figure, in its first sentence, make explicit the clinical index that is being quantified and which groups/categories are being compared.

Figure 3 does not show what the * or the added circles in the graphs mean. It must be reported if it corresponds to an indicator of statistical significance or a potential outlier.

In figure 6, the label of the axes needs to be reported. In addition, neither the test used for the correlation, nor the values ​​associated with the test are reported. It is suggested to modify the legend of the figure to facilitate the understanding of the correlation.

Discussion

General comments

The authors carry out a discussion in which they compare some of the results obtained in the study with the current body of literature, in addition to providing some concepts of potential mechanisms. However, a significant change concerning the discussion is necessary.

It is suggested to start the section by mentioning the main findings obtained in the study and then contrasting them with the evidence in the literature. Using pre-clinical models to identify potential mechanisms would add strength to this section.

Finally, it is necessary to highlight the relevance of the main findings of this study and how these findings contribute to the existing literature and improve the use of Doppler and proposed correlations in the clinic.

Specific comments:

1. In the last paragraph of discussion, this phrase “We found only a week correlation between the right and/or left UTA Doppler and the severity of UA” is not clear.

Conclusions

In this section, the authors are encouraged to highlight the study's main findings and their potential relationship with the use of steroids in high-risk pregnancies. Authors are recommended to avoid straightforwardly describing the results and make a summary that combines the findings obtained and the potential contribution of this study to the clinical practices already carried out.

Reviewer 2 Report

Well written article, highly original. This paper aims the quantification of the effect of maternal administration of dexamethasone (Dex) on the fetal circulation exclusively in l early onset preeclampsia pregnancies complicated by fetal growth restriction. Discussion and statistical analysis is good. Good work.

Author Response

We are deeply grateful to the valuable REVIEWER 2 for his very nice and honorable appreciations.

Thank you so much!

Andrei Tica,

As main correspondent author

Round 2

Reviewer 1 Report

Tica et al. report a correlation study on the variation of UA-PI, MCA-PI, CPR, and DV-PI with the response of AEDF in UA to maternal administration of dexamethasone (Dex). This study was conducted in pregnancies diagnosed with early-onset severe preeclampsia and fetal growth restriction. The authors can distinguish two AEDF groups and the effect of Dex on each variable determined via Doppler: Most with the restored absence of flow and the rest with the persistence of absent flow.

The corrections added to the manuscript by the authors have been correct. However, one last aspect must be considered.

Figure 1 and Figure 6 show the same data (for the analysis corresponding to the umbilical artery). Figure 6 only adds the data obtained for the left and right uterine arteries. It is recommended not to show the same data in two different figures.

Congratulations to the research group for the work done on this manuscript.

Author Response

We are profoundly grateful to the valuable REVIEWER 1 for his very kind appreciations!

We reiterate the thank to him for his precious time spent for reviewing two times our manuscript.

We eliminated Figure 1, so the information remained revealed only once, in Figure 5, which is more complex.